# Mixed Convective Flow of Micropolar Nanofluid across a Horizontal Cylinder in Saturated Porous Medium

**Ahmed M. Rashad** [1] , **Waqar A. Khan** [2,*] , **Saber M. M. EL-Kabeir** [1,3] and **Amal M. A. EL-Hakiem** [1]

[1] Department of Mathematics, Faculty of Science, Aswan University, Aswan 81528, Egypt; am_rashad@yahoo.com (A.M.R.); elkabeir@yahoo.com (S.M.M.E.-K.); elhakiem_amal@yahoo.com (A.M.A.E.-H.)

[2] Department of Mechanical Engineering, College of Engineering, Prince Mohammad Bin Fahd University, Al Khobar 31952, Saudi Arabia

[3] Department of Mathematics, College of Science and Humanities in Al-Kharj, Prince Sattam bin Abdulaziz University, Al-Kharj 11942, Saudi Arabia

* Correspondence: wkhan1956@gmail.com

**Abstract:** The micropolar nanofluids are the potential liquids that enhance the thermophysical features and ability of heat transportation instead of base liquids. Alumina and Titania nanoparticles are mixed in a micropolar fluid. The impact of convective boundary condition is also examined with assisting and opposing flows of both nanofluids. The main objective of this study is to investigate mixed convective flow and heat transfer of micropolar nanofluids across a cylinder in a saturated porous medium. Non-similar variables are used to make the governing equations dimensionless. The local similar and non-similar solutions are obtained by using the Runge-Kutta-Fehlberg method of seventh order. The impacts of various embedded variables on the flow and heat transfer of micropolar nanofluids are investigated and interpreted graphically. It is demonstrated that the skin friction and heat transfer rates depend on solid volume fraction of nanoparticles, Biot number, mixed convection, and material parameters.

**Keywords:** micropolar nanofluid; mixed convection; porous medium; horizontal cylinder; Convective boundary condition

## 1. Introduction

The development and evolution interest of innovative study in the area of technology science led to increasing the production of improved and modified machine tools such as synthetic lubricants and pored bearings, etc. The performance of pored bearings can be compared with the standard bearings because they have a feature like self-lubricating, which enhances their life span. The micropolar fluid is one type of fluids, which include couple stresses and the local rotating inertia. The theory of micropolar fluid is a subject of interest of several tribologists. The theory of micropolar fluid was revealed to the world by Eringen [1]. He described that the micropolar fluid has some unique advantages, such as the modeling of a wide variety of fluid flow through several media. Rashad et al. [2] reported the analytical solution for heat and mass transfer of micropolar fluid flow over a moving vertical plate in a porous medium. Li [3] derived the modified Reynolds equation for the porous media applying the Brinkman-extended Darcy equation. Rashad et al. [4] explained the heat and fluid flow characteristics of micropolar fluid flowing through a porous moving surface. They applied a uniform heat flux condition to drive the general expressions of different parameters. Isa et al. [5] and Bhatt et al. [6]

analyzed the performance of one-dimensional porous slider bearing by developing the generalized Reynolds equation in case of micropolar fluid. Rashad et al. [7] investigated the flow of micropolar fluid over a stretchable surface, and they found that the micropolar fluid-induced drag decline and enhances the wall couple's stress and the heat transfer rate of the stretched surface. Under squeeze film lubrication, Rashad et al. [8] clarified the skin-friction coefficient and heat transfer rate boosts and wall couple stress declines if the micropolar fluid vortex viscosity parameter grows in case of micropolar fluid flow on a vertical plate. Sinha et al. [9] explained that the micropolar fluid possesses enhanced viscosity especially in thin-film lubrication, which is a singular characteristic of micropolar lubricant. Rashad et al. [10] investigated the micropolar fluid flow through a horizontal circular cylinder.

On the other hand, the capability of heat transportation is required to meet the recent request of energy, and this can be achieved by employing liquid with its more impressive thermophysical features. The pioneering act on nanofluid was carried out by Choi [11]. Such type of fluids is the combination of nanoparticles in a mixture of base liquid. An augmentation in heat exchange using nanomaterials is affected by various mechanisms like sedimentation, Brownian motion, insertion of particles, transport of ballistic conduction, and thermophoresis. Nanomaterials possess fundamental features due to which it is essential in several applications regarding heat mechanisms like fuel chamber, cooling engine automobile, hybrid-electric engines, pharmacological method, boiler gas outlet, local refrigerator, temperature control, and crushing processes. Chamkha et al. [12] carried out the aspect of nanofluid mixed convective flow past a radiate cone in a porous medium. The investigations of nanofluids flow by combined convective past cylinder and sphere saturating porous medium were studied by Nazar et al. [13], Tham and Nazar [14], Tham et al. [15] and Rashad et al. [16]. Tlili et al. [17] examined the magneto-nanofluids flow past a porous stretching cylinder with multiple slips and radiation effects. Mabood et al. [18] analyzed the nanofluid forced convective flow across a circular cylinder. However, various attempts have been established about the use of nanofluids to enhance heat transfer [19–23].

Propelled by the previously mentioned investigations, the intention here is to analyze the mixed convective flow of a micropolar nanofluid past a horizontal circular cylinder in a saturating porous medium employing convective boundary conditions. The representative outcome for the velocity, temperature, friction coefficient, and Nusselt number are explaining the influences of the nanoparticles volume fraction, Biot number, mixed convection, and micropolar parameters.

## 2. Basic Equations

In this section, a model is developed to study the micropolar nanofluid flow through a horizontal circular cylinder of radius, *a* saturating non-Darcy porous medium using convective boundary conditions. The nanofluid in current exploration is composed of micropolar fluid with alumina ($Al_2O_3$) or titania ($TiO_2$). The graphical view of the problem and the coordinate system of such a flow model are exhibited in Figure 1. The flow is assisting when $T_f > T_\infty$ and opposing when $T_f < T_\infty$. The velocity of the external flow is assumed to be $\overline{u}_e(\overline{x})$, whereas the heat transfer coefficient $h_f$ is supposed to be temperature-dependent. Also, both nanoparticles and micropolar fluid are considered to be in dynamic and thermal equilibrium. These assumptions lead to the following expressions: (see Merkin [24] and Nazar et al. [25]).

$$\frac{\partial \overline{u}}{\partial \overline{x}} + \frac{\partial \overline{u}}{\partial \overline{y}} = 0 \tag{1}$$

$$\begin{aligned}\overline{u}\frac{\partial \overline{u}}{\partial \overline{x}} + \overline{v}\frac{\partial \overline{u}}{\partial \overline{y}} &= \overline{u}_e \frac{\mathrm{d}\overline{u}_e}{\mathrm{d}\overline{x}} + \frac{\mu_{nf}+\kappa}{\rho_{nf}}\frac{\partial^2 \overline{u}}{\partial \overline{y}^2} + \frac{\mu_{nf}+\kappa}{\rho_{nf}K}\left(\overline{u}_e - \overline{u}\right) \\ &\pm \frac{1}{\rho_{nf}}g^*(\rho\beta)_{nf}(T-T_\infty)\sin\left(\frac{\overline{x}}{a}\right) + \frac{\kappa}{\rho_{nf}}\frac{\partial \overline{N}}{\partial \overline{y}}\end{aligned} \tag{2}$$

$$\overline{u}\frac{\partial \overline{N}}{\partial \overline{x}} + \overline{v}\frac{\partial \overline{N}}{\partial \overline{y}} = \frac{\gamma_{nf}}{\rho_{nf}j}\frac{\partial^2 \overline{N}}{\partial \overline{y}^2} - \frac{\kappa}{\rho_{nf}j}\left(2\overline{N} + \frac{\partial \overline{u}}{\partial \overline{y}}\right) \tag{3}$$

$$\overline{u}\frac{\partial T}{\partial \overline{x}} + \overline{v}\frac{\partial T}{\partial \overline{y}} = \alpha_{nf}\frac{\partial^2 T}{\partial \overline{y}^2} \tag{4}$$

subject to the boundary conditions;

$$\bar{u} = \bar{v} = 0, \; -k_{nf}\frac{\partial T}{\partial y} = h_f(T_f - T), \; \overline{N} = 0 \text{ on } \bar{y} = 0$$

$$\bar{u} \to \bar{u}_e(\bar{x}), T \to T_\infty, \; \overline{N} \to 0 \; \text{on } \bar{y} \to \infty \tag{5}$$

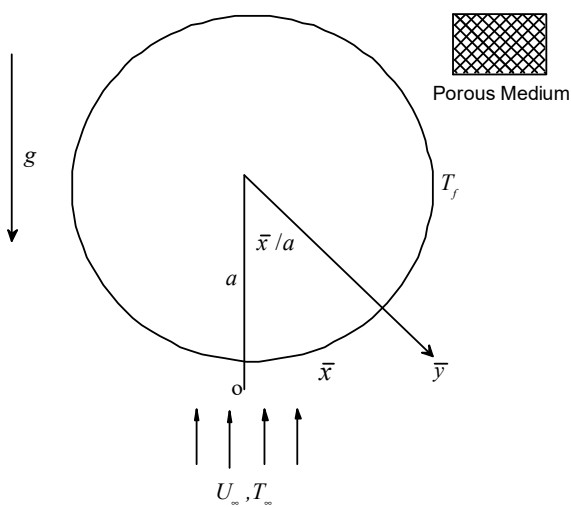

**Figure 1.** Geometry of the flow process.

The second last term on R.H.S. of Equation (2) shows the influence of the thermal buoyancy force, with positive and negative signs pertaining to the buoyancy assisting and opposing flow regimes, respectively. To obtain the non-similar result, use the following dimensionless variables;

$$\xi = \bar{x}/a, \eta = \mathrm{Re}^{1/2}(\bar{y}/a), u = \bar{u}/U_\infty, v = \mathrm{Re}^{1/2}(\bar{v}/U_\infty), u_e(\xi) = \bar{u}_e(\bar{x})/U_\infty,$$

$$\theta(\xi, \eta) = (T - T_\infty)/\left(T_f - T_\infty\right), N = a\overline{N}/U_\infty\mathrm{Re}^{1/2} \tag{6}$$

In the present work, the following thermophysical relations are utilized [26];

$$\rho_{nf} = (1-\phi)\rho_f + \phi\rho_s, \mu_{nf} = \frac{\mu_f}{(1-\phi)^{2.5}}, \alpha_{nf} = \frac{k_{nf}}{\left(\rho C_p\right)_{nf}}, (\rho\beta)_{nf} = (1-\phi)(\rho\beta)_f + \phi(\rho\beta)_s$$

$$\frac{k_{nf}}{k_f} = \frac{\left(k_s + 2k_f\right) - 2\phi\left(k_f - k_s\right)}{\left(k_s + 2k_f\right) + \phi\left(k_f - k_s\right)}, \gamma_{nf} = (\mu_{nf} + \kappa/2)j = \mu_f(\mu_{nf}/\mu_f + K/2)j$$

$$\left(\rho C_p\right)_{nf} = (1-\phi)\left(\rho C_p\right)_f + \phi\left(\rho C_p\right)_s \tag{7}$$

Here φ is nanoparticles volume fraction. The effective thermal and physical properties of nanofluids have been reported in reference [22]. Using Equations (6)–(7) in Equations (1)–(4), one can have.

$$\frac{\partial u}{\partial \xi} + \frac{\partial v}{\partial \eta} = 0 \tag{8}$$

$$u\frac{\partial u}{\partial \xi} + v\frac{\partial u}{\partial \eta} = u_e\frac{\mathrm{d}u_e}{\mathrm{d}\xi} + \frac{\rho_f}{\rho_{nf}}\left(\frac{\mu_{nf}}{\mu_f} + \Delta\right)\frac{\partial^2 u}{\partial \eta^2} + \frac{\rho_f}{\rho_{nf}}\frac{(\rho\beta)_{nf}}{(\rho\beta)_f}\lambda\theta\sin\xi$$
$$+ \frac{\rho_f}{\rho_{nf}}\left(\frac{\mu_{nf}}{\mu_f} + \Delta\right)\varepsilon(u_e - u) + \frac{\rho_f}{\rho_{nf}}\Delta\frac{\partial N}{\partial \eta} \tag{9}$$

$$u\frac{\partial N}{\partial x} + v\frac{\partial N}{\partial y} = \frac{\rho_f}{\rho_{nf}}\left(\frac{\mu_{nf}}{\mu_f} + \frac{\Delta}{2}\right)\frac{\partial^2 N}{\partial y^2} - \frac{\rho_{nf}}{\rho_{nf}}\Delta B\left(2N + \frac{\partial u}{\partial y}\right) \tag{10}$$

$$u\frac{\partial\theta}{\partial\xi} + v\frac{\partial\theta}{\partial\eta} = \frac{1}{\mathrm{Pr}}\frac{\alpha_{nf}}{\alpha_f}\frac{\partial^2\theta}{\partial\eta^2} \tag{11}$$

The dimensionless boundary conditions become:

$$u = v = 0,\ \frac{k_{nf}}{k_f}\frac{\partial\theta}{\partial\eta} = -Bi[1-\theta],\ S = 1 \text{ on } \eta = 0$$

$$u \to u_e,\ \theta \to 0,\ S \to 0 \text{ as } \eta \to \infty \tag{12}$$

where $\lambda$ is the mixed convection or buoyancy parameter, Gr is the Grashof number, $\varepsilon$ is the permeability parameter (porous medium parameter), $\Delta$ and $B$ are the material parameters, Bi is the Biot number, and Pr is the Prandtl number. These dimensionless numbers can be written as;

$$\lambda = \frac{g\beta_f(T_f - T_\infty)a}{U_\infty^2} = \pm\frac{Gr}{\mathrm{Re}^2}, Gr = \frac{g\beta_f(T_f - T_\infty)a^3}{v_f^2}, \varepsilon = (v_f a)/(U_\infty K), B = \frac{av_f}{jU_\infty}, \Delta = \frac{\kappa}{\mu_f}$$

$$\mathrm{Pr} = \frac{v_f}{\alpha_f}, Bi = \frac{h_f a}{\mathrm{Re}^{1/2}k_f} \tag{13}$$

It is important to note that $\lambda > 0$ and $\lambda < 0$ show the assisting and opposing flows, respectively.

According to [25], the stream function $\psi$ can be defined as $u = \partial\psi/\partial\eta$, $v = -\partial\psi/\partial\xi$ and then, Equation (8) is satisfied. Using the following transformations.

$$\psi = \xi f(\xi,\eta),\ \theta = \theta(\xi,\eta) \tag{14}$$

Into Equations (8)–(10), we get:

$$\frac{\rho_f}{\rho_{nf}}\left(\frac{\mu_{nf}}{\mu_f} + \Delta\right)f''' + ff'' - f'^2 + \frac{\sin\xi\cos\xi}{\xi} + \lambda\frac{\sin\xi}{\xi}\frac{\rho_f}{\rho_{nf}}\frac{(\rho\beta)_{nf}}{(\rho\beta)_f}\theta$$
$$+ \frac{\rho_f}{\rho_{nf}}\left(\frac{\mu_{nf}}{\mu_f} + \Delta\right)\varepsilon\left(\frac{\sin\xi}{\xi} - f'\right) + \frac{\rho_f}{\rho_{nf}}\Delta g' = \xi\left(f'\frac{\partial f'}{\partial\xi} - f''\frac{\partial f}{\partial\xi}\right) \tag{15}$$

$$\frac{\rho_f}{\rho_{nf}}\left(\frac{\mu_{nf}}{\mu_f} + \frac{\Delta}{2}\right)g'' + fg' - f'g - \frac{\rho_{nf}}{\rho_{nf}}\Delta B(2g + f'') = \xi\left(f'\frac{\partial g}{\partial\xi} - g'\frac{\partial f}{\partial\xi}\right) \tag{16}$$

$$\frac{1}{\mathrm{Pr}}\frac{\alpha_{nf}}{\alpha_f}\theta'' + f\theta' = \xi\left(f'\frac{\partial\theta}{\partial\xi} - \theta'\frac{\partial f}{\partial\xi}\right) \tag{17}$$

The accompanying non-dimensional boundary conditions become:

$$f = f' = 0, g = 0, \frac{k_{nf}}{k_f}\theta' = -Bi[1-\theta] \quad \text{at } \eta = 0$$
$$f' \to \frac{\sin\xi}{\xi}, g = 0, \theta \to 0 \quad \text{as} \quad \eta \to \infty \tag{18}$$

Finally, the expressions for the drag friction $C_f$ and local Nusselt number ($Nu$) can be written as:

$$\begin{aligned}C_f(\xi) &= \mathrm{Re}^{1/2}\frac{\tau_w}{\rho_f U_\infty^2} = \mathrm{Re}^{1/2}\frac{((\mu_{nf}+\kappa)\partial\overline{u}/\partial\overline{y}+\kappa\overline{N})_{\overline{y}=0}}{\rho_f U_\infty^2}\\ &= \xi\left(\left(\frac{\mu_{nf}}{\mu_f} + \Delta\right)f''(\xi,0) + \Delta g(\xi,0)\right)\end{aligned} \tag{19}$$

$$Nu(\xi) = \text{Re}^{-1/2}\frac{q_w a}{k_f(T_f - T_\infty)} = \text{Re}^{-1/2}\frac{a(\partial T/\partial \overline{y})_{\overline{y}=0}}{k_f(T_f - T_\infty)} = -\frac{k_{nf}}{k_f}\theta'(\xi, 0) \tag{20}$$

## 3. Numerical Method

### 3.1. First Level of Truncation (Local Similarity Solution)

Following Sparrow et al. [22,23], for the first level of truncation, it is assumed that the zeta derivatives in Equations (15)–(17) are neglected, and the equations can be rewritten as:

$$\frac{\rho_f}{\rho_{nf}}\left(\frac{\mu_{nf}}{\mu_f} + \Delta\right)f''' + ff'' - f'^2 + \frac{\sin\xi\cos\xi}{\xi} + \lambda\frac{\sin\xi}{\xi}\frac{\rho_f}{\rho_{nf}}\frac{(\rho\beta)_{nf}}{(\rho\beta)_f}\theta$$
$$+ \frac{\rho_f}{\rho_{nf}}\left(\frac{\mu_{nf}}{\mu_f} + \Delta\right)\varepsilon\left(\frac{\sin\xi}{\xi} - f'\right) + \frac{\rho_f}{\rho_{nf}}\Delta g' = 0 \tag{21}$$

$$\frac{\rho_f}{\rho_{nf}}\left(\frac{\mu_{nf}}{\mu_f} + \frac{\Delta}{2}\right)g'' + fg' - f'g - \frac{\rho_{nf}}{\rho_{nf}}\Delta B(2g + f'') = 0 \tag{22}$$

$$\frac{1}{\text{Pr}}\frac{\alpha_{nf}}{\alpha_f}\theta'' + f\theta' = 0 \tag{23}$$

Subject to the same boundary conditions (18).

$$f = f' = 0, g = 0, \frac{k_{nf}}{k_f}\theta' = -Bi[1 - \theta] \quad \text{at} \quad \eta = 0$$
$$f' \to \frac{\sin\xi}{\xi}, g = 0, \theta \to 0 \quad \text{as} \quad \eta \to \infty \tag{24}$$

Equations (23)–(25) are solved numerically with the boundary conditions (26), and similar solutions are obtained.

### 3.2. Second Level of Truncation (Local Non-Similarity)

Following Sparrow and co-workers [22,23], all the terms are retained in Equations (15)–(17) without any approximation and new auxiliary functions $F(\xi, \eta), G(\xi, \eta)$ and $\Theta(\xi, \eta)$ are assumed which are defined by:

$$F = \frac{\partial f}{\partial \xi}, G = \frac{\partial g}{\partial \xi}, \Theta = \frac{\partial \theta}{\partial \xi}$$

Using these functions, Equations (15)–(17) can be re-written as:

$$\frac{\rho_f}{\rho_{nf}}\left(\frac{\mu_{nf}}{\mu_f} + \Delta\right)f''' + ff'' - f'^2 + \frac{\sin\xi\cos\xi}{\xi} + \lambda\frac{\sin\xi}{\xi}\frac{\rho_f}{\rho_{nf}}\frac{(\rho\beta)_{nf}}{(\rho\beta)_f}\theta$$
$$+ \frac{\rho_f}{\rho_{nf}}\left(\frac{\mu_{nf}}{\mu_f} + \Delta\right)\varepsilon\left(\frac{\sin\xi}{\xi} - f'\right) + \frac{\rho_f}{\rho_{nf}}\Delta g' - \xi(f'F' - f''F) = 0 \tag{25}$$

$$\frac{\rho_f}{\rho_{nf}}\left(\frac{\mu_{nf}}{\mu_f} + \frac{\Delta}{2}\right)g'' + fg' - f'g - \frac{\rho_{nf}}{\rho_{nf}}\Delta B(2g + f'') - \xi(f'G - Fg') = 0 \tag{26}$$

$$\frac{1}{\text{Pr}}\frac{\alpha_{nf}}{\alpha_f}\Theta'' + f\theta' - \xi(f'\Theta - \theta'F) = 0 \tag{27}$$

With the same boundary conditions (18).

Differentiating Equations (25)–(27) and boundary conditions (18) with respect to $\xi$ and neglecting $\xi$ derivatives again, we get:

$$\left.\begin{array}{l} \dfrac{\rho_f}{\rho_{nf}}\left(\dfrac{\mu_{nf}}{\mu_f}+\Delta\right)F''' + Ff'' + fF'' - 2f'F' + \dfrac{\cos^2\xi-\sin^2\xi}{\xi} - \dfrac{\cos\xi-\sin\xi}{\xi^2} + \\[2mm] \lambda\dfrac{\cos\xi}{\xi}\dfrac{\rho_f}{\rho_{nf}}\dfrac{(\rho\beta)_{nf}}{(\rho\beta)_f}\theta - \lambda\dfrac{\sin\xi}{\xi^2}\dfrac{\rho_f}{\rho_{nf}}\dfrac{(\rho\beta)_{nf}}{(\rho\beta)_f}\theta + \lambda\dfrac{\sin\xi}{\xi}\dfrac{\rho_f}{\rho_{nf}}\dfrac{(\rho\beta)_{nf}}{(\rho\beta)_f}\Theta + \\[2mm] \dfrac{\rho_f}{\rho_{nf}}\left(\dfrac{\mu_{nf}}{\mu_f}+\Delta\right)\varepsilon\left(\dfrac{\cos\xi}{\xi} - \dfrac{\sin\xi}{\xi^2} - F'\right) + \dfrac{\rho_f}{\rho_{nf}}\Delta G' - \xi\left(F'^2 - F''F\right) = 0 \end{array}\right\} \tag{28}$$

$$\frac{\rho_f}{\rho_{nf}}\left(\frac{\mu_{nf}}{\mu_f}+\frac{\Delta}{2}\right)G'' + 2Fg' + fG' - F'g - 2f'G - \frac{\rho_{nf}}{\rho_{nf}}\Delta B(2G+F'') - \xi(F'G - FG') = 0 \tag{29}$$

$$\frac{1}{Pr}\frac{\alpha_{nf}}{\alpha_f}\Theta'' + 2F\theta' + f\Theta' - f'\Theta - \xi(F'\Theta - \Theta'F) = 0 \tag{30}$$

With additional boundary conditions.

$$\left.\begin{array}{l} F = F' = 0, G = 0, \dfrac{k_{nf}}{k_f}\Theta'(0) = Bi[\Theta(0)] \text{ at } \eta = 0 \\[2mm] F' = \dfrac{\cos\xi\sin\xi}{\xi^2}, G = 0, \Theta \to 0 \text{ as } \eta \to \infty \end{array}\right\} \tag{31}$$

The system of nonlinear ordinary differential Equations (28)–(30) with boundary conditions (31) can be solved numerically using the Runge-Kutta-Fehlberg method of seventh order. In this method, twelve evaluations are found to be enough for each step. For higher accuracy, this method provides the most efficient results. The step size $\Delta\eta = 0.001$ and a convergence criterion of $10^{-6}$ were used in the numerical computations. The asymptotic boundary conditions, given by Equations (31), were replaced by using a value of 12 for the similarity variable $\eta_{max}$ as follows.

$$F'(\xi, 12) = 0, \Theta(\xi, 12) = 0, G(\xi, 12) = 0 \tag{32}$$

The selection of $\eta_{max} = 12$ guarantees that all numerical results approached the asymptotic values accurately. For further details, readers are advised to see refs [27,28].

## 4. Results and Discussions

The mixed convective flow of micropolar fluids with alumina and Titania nanoparticles across a horizontal circular cylinder is investigated in a saturated porous medium. The effects of governing parameters with the convective boundary condition are analyzed for assisting and opposing flows. Both local similarity and non-similarity equations with the selected boundary conditions are solved numerically. The results of the local Nusselt number are compared with the existing results in Table 1 and are found in excellent agreement.

**Table 1.** Comparison of $-\theta'(0)$ for different values of $\lambda$ for $\varepsilon = \varphi = \Delta = 0$, $Bi \to \infty$ and $Pr = 1.0$.

| $\lambda$ | Merkin [24] | Nazar et al. [25] | Present Results |
|:---:|:---:|:---:|:---:|
| −1.75 | 0.4199 | 0.4205 | 0.4159 |
| −1.5 | 0.4576 | 0.4601 | 0.4589 |
| −1.0 | 0.5067 | 0.5080 | 0.5060 |
| −0.5 | 0.5420 | 0.5430 | 0.5406 |
| 0.0 | 0.5705 | 0.5710 | 0.5700 |
| 0.5 | 0.5943 | 0.5949 | 0.5923 |
| 0.88 | 0.6096 | 0.6112 | 0.6054 |
| 0.89 | 0.6110 | 0.6116 | 0.6103 |
| 1.0 | 0.6158 | 0.6160 | 0.6154 |
| 2.0 | 0.6497 | 0.6518 | 0.6468 |
| 5.0 | 0.7315 | 0.7320 | 0.7304 |

The effects of buoyancy and material parameters on the dimensionless velocity are presented in Figures 2 and 3 for (a) assisting and (b) opposing flows, respectively. The other parameters are kept constant. In the case of assisting flow, the dimensionless velocity of both $Al_2O_3$ and $TiO_2$ base micropolar fluids increases within the hydrodynamic boundary layer with a mixed convection parameter, see Figure 2a,b. The velocity satisfies both boundary conditions. On the other side, the material parameter $\Delta$ reduces the dimensionless velocity. This is because the velocity of micropolar fluids is coupled with the macroscopic rotation of the fluid particles. For both micropolar nanofluids, we observe the same behavior. For opposing flows, the dimensionless velocity shows opposite behavior, Figure 3a,b. It decreases with the mixed convection and the material parameters inside the boundary layer. For smaller values of both parameters, the dimensionless velocity increases from the surface to the boundary layer. However, for larger values of both parameters, the velocity decreases close to the surface and then increases and converges.

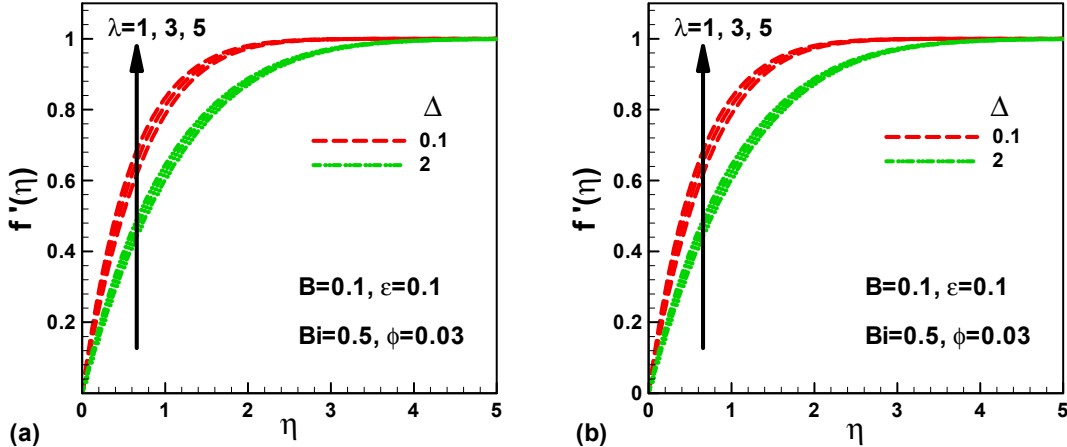

**Figure 2.** Variation of dimensionless velocity with $\Delta$ and $\lambda$ for assisting flow of micropolar fluid with (**a**) $Al_2O_3$ and (**b**) $TiO_2$ nanoparticles.

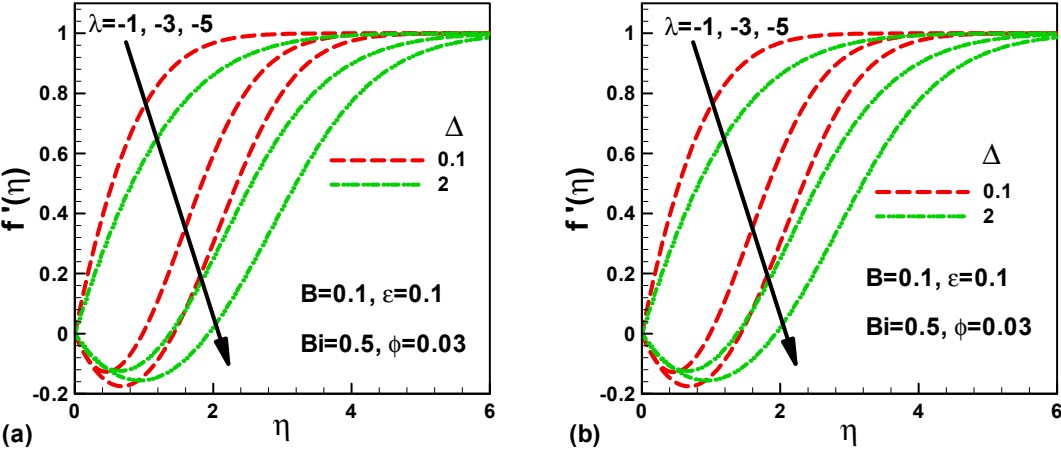

**Figure 3.** Variation of dimensionless velocity with $\Delta$ and $\lambda$ for opposing flow of micropolar fluid with (**a**) $Al_2O_3$ and (**b**) $TiO_2$ nanoparticles.

The effects of the same parameters on the microrotation velocity are depicted in Figure 4 for assisting flows and in Figure 5 for opposing flows of micropolar nanofluids. In the hydrodynamic boundary layer, the variation of microrotation velocity with the mixed convection and material parameters is negligible for both nanofluids, Figure 4a,b. For smaller values of the material parameter, the microrotation velocity is almost zero in both cases. In fact, both velocity and temperature fields become independent of the microstructure of the fluid. However, as the values of the material

parameter increase, the velocity goes in the negative direction and then approaches the boundary layer. For smaller values of $\Delta$, the microrotation velocity is also negligible in case of opposing flows for both nanofluids, Figure 5a,b. However, for large values of $\Delta$, the microrotation velocity shows oscillatory behavior in both cases.

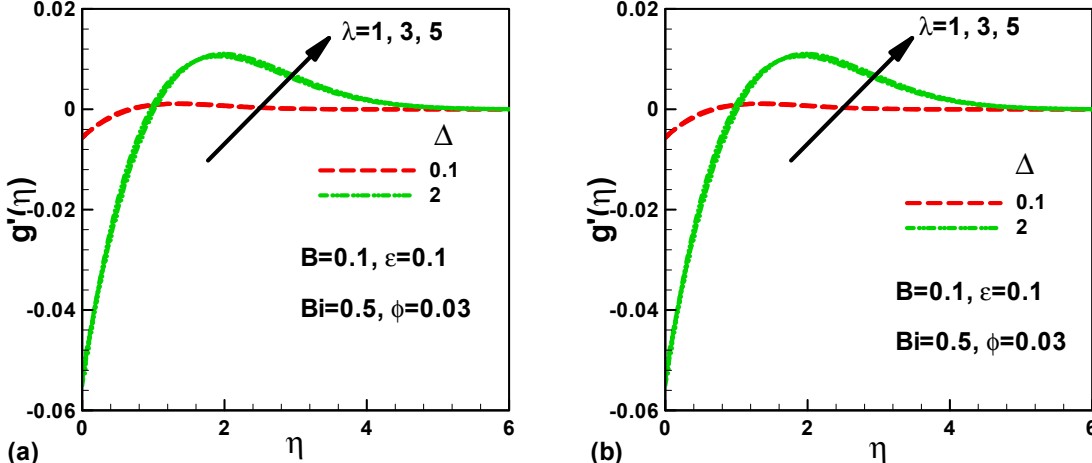

**Figure 4.** Variation of dimensionless rotational velocity with $\Delta$ and $\lambda$ for assisting flow of micropolar fluid with (**a**) $Al_2O_3$ and (**b**) $TiO_2$ nanoparticles.

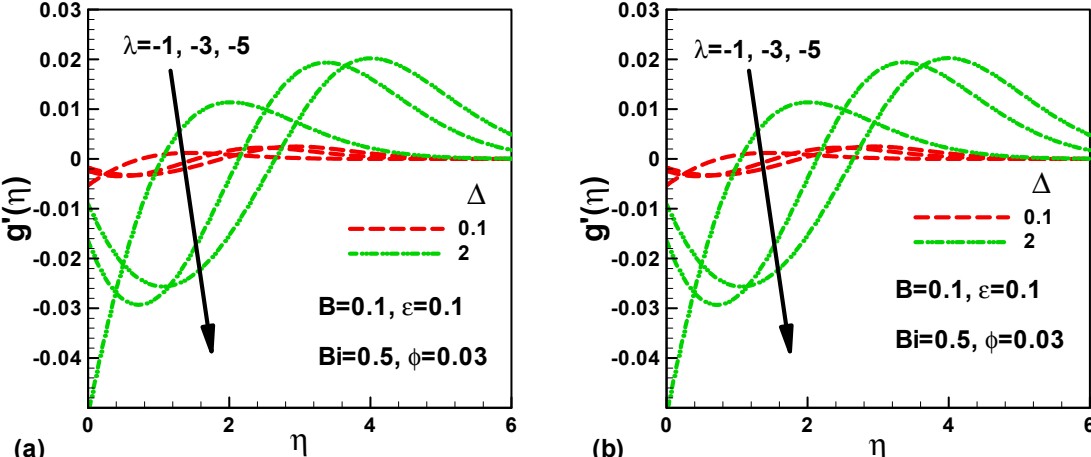

**Figure 5.** Variation of dimensionless rotational velocity with $\Delta$ and $\lambda$ for opposing flow of micropolar fluid with (**a**) $Al_2O_3$ and (**b**) $TiO_2$ nanoparticles.

The impacts of mixed convection and material parameters on the dimensionless temperature are presented in Figure 6 for assisting flows and in Figure 7 for opposing flows of micropolar nanofluids. In the thermal boundary layer, the dimensionless temperature decreases with both parameters for both nanofluids, Figure 6a,b, respectively. In both cases, the temperature at the surface is highest and decreases to ambient temperature. The boundary layer thickness is found to be the same for both nanofluids. It is important to note that the dimensionless temperature is almost independent of the mixed convection parameter. For opposing flows of micropolar nanofluids, the dimensionless temperature increases with $\lambda$ and decreases with $\Delta$ in both cases, Figure 7a,b. However, this difference increases with a decrease in $\lambda$. The surface temperature is highest and decreases to ambient temperature. The rate of convergence increases with a decrease in $\lambda$ for both cases.

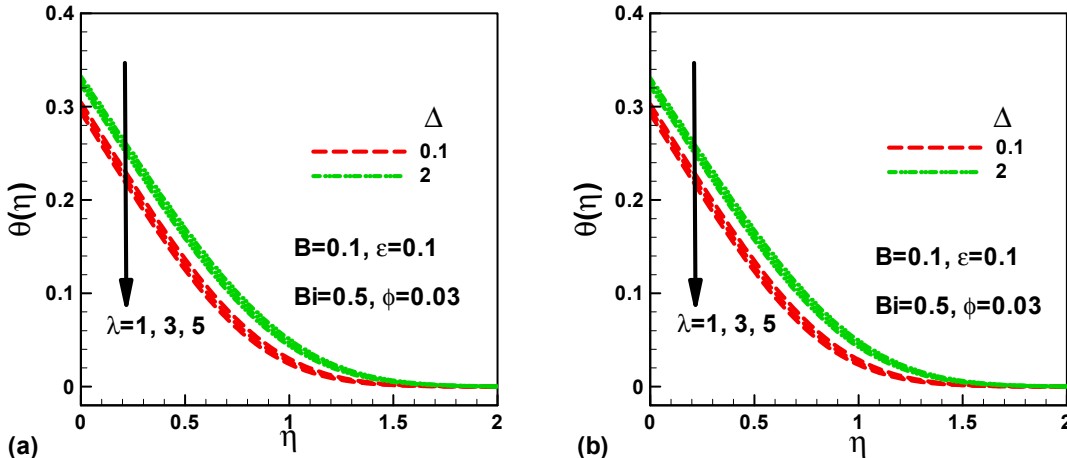

**Figure 6.** Variation of dimensionless temperature with $\Delta$ and $\lambda$ for assisting flow of micropolar fluid with (**a**) $Al_2O_3$ and (**b**) $TiO_2$ nanoparticles.

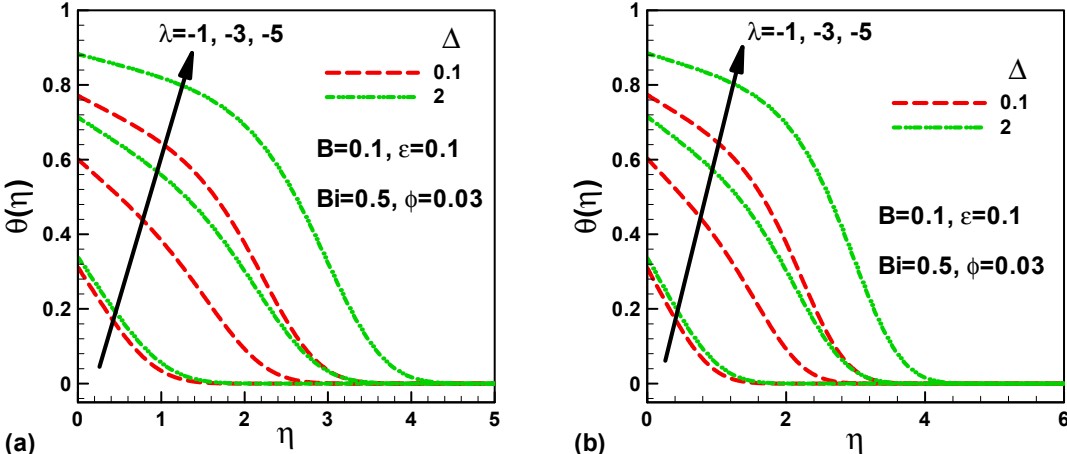

**Figure 7.** Variation of dimensionless temperature with $\Delta$ and $\lambda$ for opposing flow of micropolar fluid with (**a**) $Al_2O_3$ and (**b**) $TiO_2$ nanoparticles.

The variation of skin friction along the surface with $\Delta$ and $\lambda$ is depicted in Figure 8a for assisting flow and in Figure 8b for opposing the flow of micropolar-$Al_2O_3$ nanofluids. As expected, the friction is zero at the front stagnation point and increases along the surface up to maximum and then decreases up to the rear stagnation point for both flows. In assisting flows, the maximum friction increases with both mixed convection and material parameters. However, in opposing flows, the maximum friction decreases with increasing mixed convection parameter. The material parameter does not affect the opposing flow. The same effects of mixed convection and material parameters on the skin friction can be observed in Figure 9a for assisting flow and in Figure 9b for opposing flow of micropolar-$TiO_2$ nanofluid. Due to higher density of $TiO_2$ nanoparticles, the skin friction for $TiO_2$ nanofluid is little bit higher than $Al_2O_3$ nanofluid. The effects of solid volume fraction of $TiO_2$ nanoparticles and Biot number on the skin friction are presented in Figure 10a for assisting flow and in Figure 10b for opposing flow of micropolar nanofluids. The skin friction at the front stagnation point is zero and increases up to the maximum along the cylinder surface and then decreases up to the rear stagnation point in case of assisting flow. However, in opposing flow, skin friction becomes zero before the rear stagnation point. As expected, the skin friction increases with the solid volume fraction of $TiO_2$ nanoparticles due to the higher density of nanoparticles (Table 2). The small values of Biot number generate small temperature gradients within the cylinder, and the temperature gradients increase with an increase in

the Biot number. The same behavior can be observed for the opposing flows. However, in this case, the maximum skin friction is found to be less than the assisting flows.

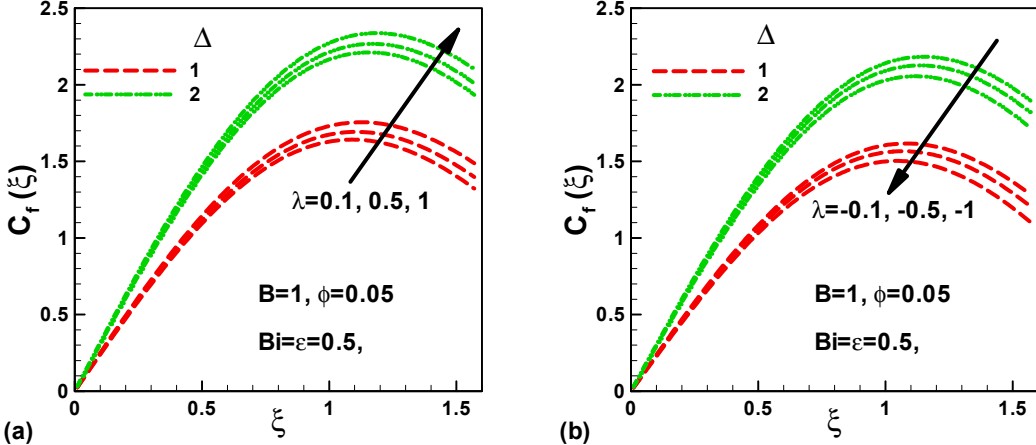

**Figure 8.** Variation of skin friction along the surface with $\Delta$ and $\lambda$ for (**a**) assisting flow and (**b**) opposing flow of micropolar-Al$_2$O$_3$ nanofluids.

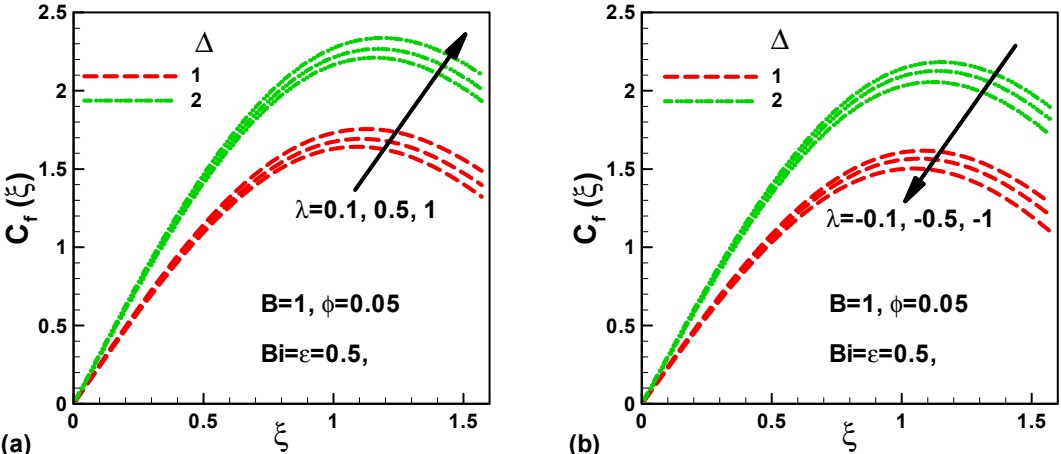

**Figure 9.** Variation of skin friction along the surface with solid volume fraction of nanoparticles and Biot number for (**a**) assisting flow and (**b**) opposing flow of micropolar-TiO$_2$ nanofluids.

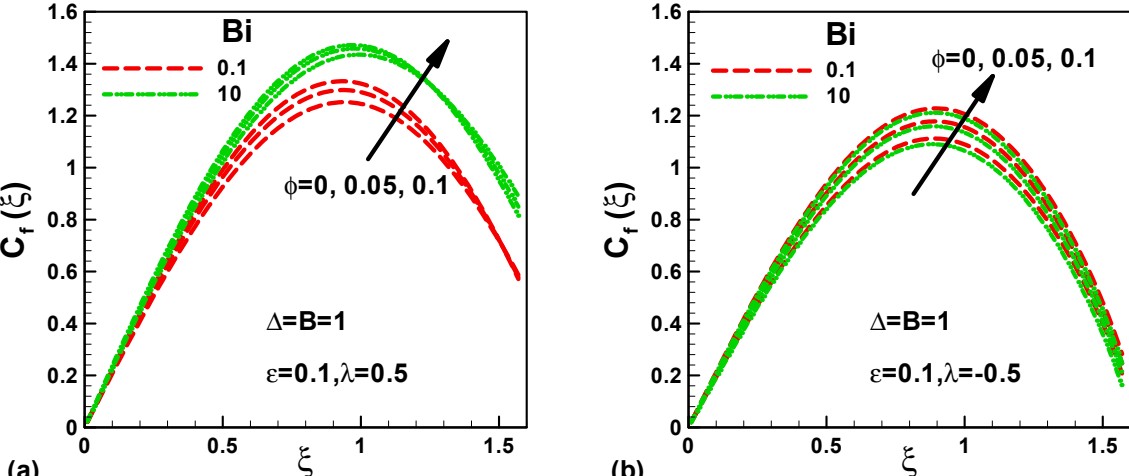

**Figure 10.** Variation of skin friction along the surface with solid volume fraction of nanoparticles and Biot number for (**a**) assisting flow and (**b**) opposing flow of micropolar-TiO$_2$ nanofluids.

**Table 2.** Thermo-physical properties of water and nanoparticles (see Oztop and Abu-Nada [22]).

| Property | Pure Water | Alumina $Al_2O_3$ | Titanium Oxide ($TiO_2$) |
|---|---|---|---|
| $\rho$ (kg m$^{-3}$) | 997.1 | 3970 | 4250 |
| $C_P$ (J kg$^{-1}$ K$^{-1}$) | 4179 | 765 | 686.2 |
| k (W m$^{-1}$ K$^{-1}$) | 0.613 | 40 | 8.9538 |
| $\beta$ (K$^{-1}$) | $21 \times 10^{-5}$ | $0.85 \times 10^{-5}$ | $0.9 \times 10^{-5}$ |

The variation of Nusselt number along the cylinder surface with mixed convection and material parameters is illustrated in Figure 11a for assisting flow and in Figure 11b for opposing the flow of micropolar-$Al_2O_3$ nanofluids. In both cases, the local Nusselt number is found to be maximum at the front stagnation point and decreases along the surface towards the rear stagnation point. In case of assisting flow, the local Nusselt numbers are found to be higher than the opposing flow. In both cases, the local Nusselt number decreases with increasing material parameters. However, the mixed convection parameters have opposite effect on both flows, see Figure 11a,b. The variation of local Nusselt number along the surface with solid volume fraction of nanoparticles and Biot number is demonstrated in Figure 12a for assisting flow and in Figure 12b for opposing flow of micropolar-$TiO_2$ nanofluids. Again, the maximum local Nusselt number is found to be at the front stagnation point decreases along the surface. In both cases, the local Nusselt number increases with an increase in the solid volume fraction of $TiO_2$ nanoparticles and Biot number.

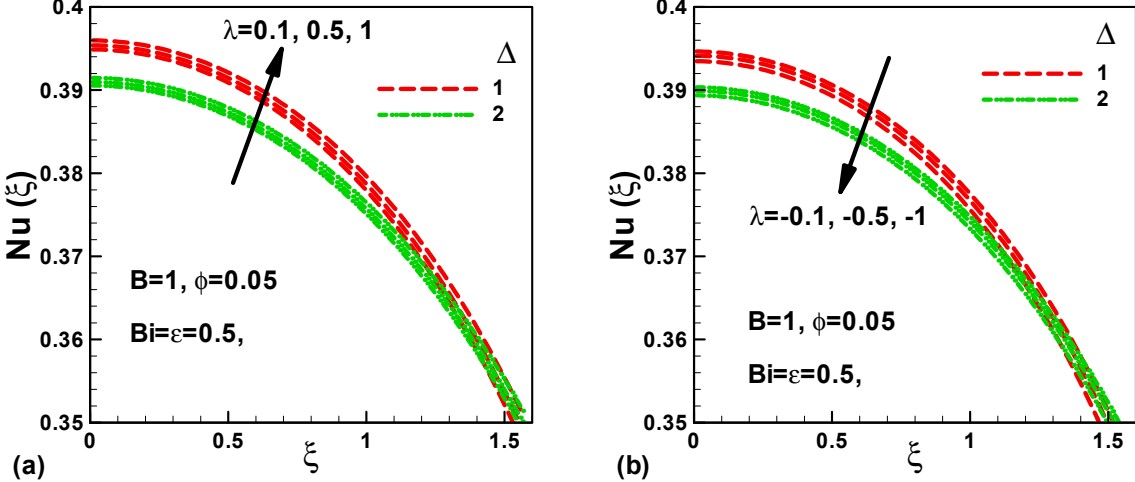

**Figure 11.** Variation of local Nusselt number along the surface with $\Delta$ and $\lambda$ for (**a**) assisting flow and (**b**) opposing flow of micropolar-$Al_2O_3$ nanofluids.

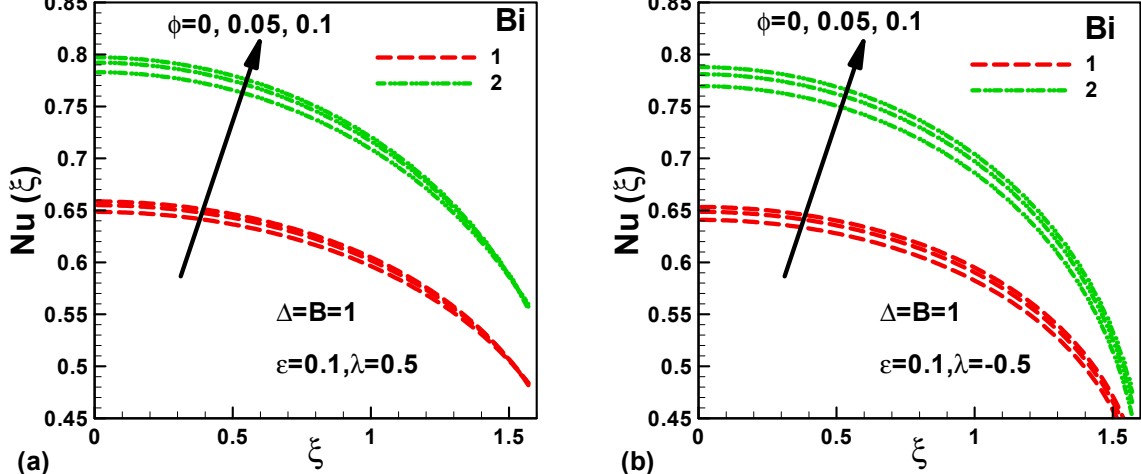

**Figure 12.** Variation of local Nusselt number along the surface with solid volume fraction of nanoparticles and Biot number for (**a**) assisting flow and (**b**) opposing flow of micropolar-TiO$_2$ nanofluids.

## 5. Conclusions

The following conclusions were drawn from this study:

1.  For opposing flows, the dimensionless velocity decreases with the mixed convection and material parameters inside the boundary layer;
2.  The smaller values of $\Delta$ show no effect on the microrotation velocity. However, for large values of $\Delta$, the microrotation velocity shows oscillatory behavior;
3.  For opposing flows of micropolar nanofluids, the dimensionless temperature increases with $\lambda$ and decreases with $\Delta$;
4.  In assisting flows, the maximum friction increases with both mixed convection and material parameters;
5.  The skin friction at the front stagnation point is zero and increases up to the maximum along the cylinder surface;
6.  In the case of assisting flow, the local Nusselt numbers are found to be higher than the opposing flow;
7.  The local Nusselt number increases with an increase in the solid volume fraction of TiO$_2$ nanoparticles and Biot number.

However, employing non-conventional fluids in the process of cooling is one of the most crucial study fields due to the importance of the coolant kind on the quality of the final product and the mechanical properties of the surfaces that have to be cooled.

**Author Contributions:** "conceptualization, A.M.R. and W.A.K.; methodology, W.A.K.; software, S.M.M.E.-K.; validation, A.M.A.E.-H., S.M.M.E.-K. and W.A.K.; formal analysis, S.M.M.E.-K.; investigation, A.M.R.; resources, W.A.K.; data curation, S.M.M.E.-K.; writing—original draft preparation, A.M.A.E.-H.; writing—review and editing, A.M.R.; visualization, A.M.R.; supervision, W.A.K.; project administration, S.M.M.E.-K.; funding acquisition, S.M.M.E.-K.".

**Funding:** This research received no external funding.

**Conflicts of Interest:** The authors declare no conflict of interest.

## Nomenclature

| | |
|---|---|
| B | Material parameter |
| Bi | Biot number |
| $C_p$ | Specific heat at constant pressure |
| $C_f$ | Skin-friction coefficient |
| $f$ | Dimensionless stream function |
| $f'$ | Dimensionless velocity |
| $g$ | Acceleration due to gravity |
| $Gr_x$ | Local Grash of number |
| $h_f$ | Convective heat transfer coefficient |
| $k$ | Thermal conductivity |
| $K$ | Permeability of porous medium |
| $Nu$ | Local Nusselt number |
| Pr | Prandtl number |
| $T$ | Temperature of the fluid in the boundary layer |
| $T_\infty$ | Temperature of the ambient fluid |
| $T_f$ | Uniform temperature of the cylinder surface |
| $\overline{u}_e(\overline{x})$ | Velocity of the external flow |
| $U\infty$ | Free stream velocity |
| $u, v$ | Dimensionless fluid velocities in the $\xi, \eta$ directions |
| $\overline{u}, \overline{v}$ | Dimensional fluid velocities in the $\overline{x}, \overline{y}$ directions |
| $\overline{x}, \overline{y}$ | Dimensional axes in the direction along and normal to the surface |
| $\beta$ | Volumetric coefficient of thermal expansion |
| $\psi$ | Stream function |
| $\varepsilon$ | Permeability parameter |
| $\lambda$ | Mixed convection parameter |
| $\rho$ | Fluid density |
| $\mu$ | Dynamic viscosity of the fluid |
| $\nu$ | Kinematic viscosity of the fluid |
| $\theta$ | Dimensionless temperature function |
| $\phi$ | Nanoparticles volume fraction |
| $\xi$ | Dimensionless coordinate |
| $\eta$ | pseudo-similarity variable |
| $\Delta$ | Dimensionless material parameter |

**Subscripts**

| | |
|---|---|
| $f$ | Pure fluid |
| $nf$ | nanofluid |
| $s$ | Solid nanoparticle |

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
