# Peer review of "Mixed Convective Flow of Micropolar Nanofluid across a Horizontal Cylinder in Saturated Porous Medium"

_applsci, doi:10.3390/app9235241_

Round 1

Reviewer 1 Report

The manuscript investigates convective flow of a micropolar nanofluid across a horizontal circular cylinder in a saturated porous medium. Numerical resolution of the governing differential equations are looked for by means of Runge-Kutta-Fehlberg method of seventh order, and the results are presented in terms of characteristic non-dimensional quantities.

The work seems original and of interest to many researchers, especially in the context of increasing importance of mechanics of continua in nano-scale. The manuscript is well-organized and provides sufficient background of the problem; and in the reviewer's opinion, worth publishing on Applied Sciences journal after minor corrections listed below.

In Eqs.1-4, please check if x  and   y  should be replaced by  \bar x  and \bar y , as Fig.1 and nomenclature suggest. A hint on finding the nomenclature at the end of the paper might help the readers. A brief discussion on the selection of this particular numerical method would be good. What are other possibilities? What is the main advantage of this particular method over finite volume techniques or collocation-based methods? The reviewer suggests addition of a couple of references to support the authors' choice. Most of the comments on numerical results are of reporting type. Brief physical reasoning of the numerical results would increasethe quality of the paper. Please highlight the difference between effects of Alumina and Titania nanoparticles on representive quantities of the problem. Please provide some hints on the importance of the main findings of this study on possible engineering applications. Please highlight main differences/novelty of this work with respect to [25].

Author Response

Reviewer 1: Comments and Suggestions for Authors

The manuscript investigates convective flow of a micropolar nanofluid across a horizontal circular cylinder in a saturated porous medium. Numerical resolution of the governing differential equations are looked for by means of Runge-Kutta-Fehlberg method of seventh order, and the results are presented in terms of characteristic non-dimensional quantities.

The work seems original and of interest to many researchers, especially in the context of increasing importance of mechanics of continua in nano-scale. The manuscript is well-organized and provides sufficient background of the problem; and in the reviewer's opinion, worth publishing on Applied Sciences journal after minor corrections listed below.

In Eqs.1-4, please check if x and y should be replaced by  \bar x  and \bar y , as Fig.1 and nomenclature suggest. A hint on finding the nomenclature at the end of the paper might help the readers.

Response:  Corrected accordingly.

A brief discussion on the selection of this particular numerical method would be good. What are other possibilities? What is the main advantage of this particular method over finite volume techniques or collocation-based methods? The reviewer suggests addition of a couple of references to support the authors' choice. Most of the comments on numerical results are of reporting type.

Response:  We followed non-similar method given by Sparrow et al. [22, 23]. This method is considered as the standard method for non-similar equations.

Brief physical reasoning of the numerical results would increase the quality of the paper. Please highlight the difference between effects of Alumina and Titania nanoparticles on representative quantities of the problem.

Response:  We agreed with the reviewer. The physical reasoning of the numerical results has been explained in the results and discussion section.

Please provide some hints on the importance of the main findings of this study on possible engineering applications.

Response:  Done accordingly, please see the final paragraph in the conclusion section.

Please highlight the main differences/novelty of this work with respect to [25]. 

Response:  In [25], Nazar et al. examined the mixed convection boundary-layer flow from a horizontal circular cylinder in micropolar fluids. Keeping fixed all thermal and physical properties of the nanoparticles, we are considering a nanofluid in our investigation. We intend to combine the micropolar flow theory with water-based nanofluid flowing across a horizontal circular cylinder embedded in a porous medium in the presence of convective boundary conditions. We focused on velocity, temperature distribution, micro-rotation components, skin friction coefficients and heat transfer influenced by nanoparticles volume fraction, Biot number and mixed convection parameter.

with the advent of nanofluids, the heat transfer capabilities of conventional fluids are enhanced dramatically, resulting in the broader use of such fluids in many engineering applications as well as in the fields of physics, magnetic drug targeting, solar collectors

Reviewer 2 Report

The authors examined the factors that affected mixed convective flow of micropolar nano fluids with alumina and titania across a cylinder in a saturated porous medium. Overall, this is a well-written manuscript. The authors stated the research problems clearly in the field and had a solid foundation for their study objectives. All equations were correctly defined and derived properly, and all graphs in the Results section were clearly presented, which supports the manuscript's conclusion.

The reviewer thought the improvement could be made in the abstract writing. The authors started the abstract with the study objectives which feels a little out of place. If the authors laid the background and problem first, it would have been more convincing why they wanted to study the flows of nano fluids and what they wanted to accomplish.

There are some places in the text that need attention. For example:

grammar/punctuation issues in Line 27, 28, 49, 62, 75, 100, 108, and 110. text editing in Line 123 clarification in Line 145. It was a bit confusing when you have both "a" and "b" standing for different things. Line 94, "reference" and "equations" should be added before the reference number and the equation number in the text format of references section ( they don't match with each other) font of Section 2 heading (should be capitalized)

Author Response

(The authors gave the same response as above.)
